# MSFragger-DDA+ enhances peptide identification sensitivity with full isolation window search

Fengchao Yu [1] ✉, Yamei Deng [1] & Alexey I. Nesvizhskii [1,2] ✉

Liquid chromatography-mass spectrometry based proteomics, particularly in the bottom-up approach, relies on the digestion of proteins into peptides for subsequent separation and analysis. The most prevalent method for identifying peptides from data-dependent acquisition mass spectrometry data is database search. Traditional tools typically focus on identifying a single peptide per tandem mass spectrum, often neglecting the frequent occurrence of peptide co-fragmentations leading to chimeric spectra. Here, we introduce MSFragger-DDA+, a database search algorithm that enhances peptide identification by detecting co-fragmented peptides with high sensitivity and speed. Utilizing MSFragger's fragment ion indexing algorithm, MSFragger-DDA+ performs a comprehensive search within the full isolation window for each tandem mass spectrum, followed by robust feature detection, filtering, and rescoring procedures to refine search results. Evaluation against established tools across diverse datasets demonstrated that, integrated within the Frag-Pipe computational platform, MSFragger-DDA+ significantly increases identification sensitivity while maintaining stringent false discovery rate control. It is also uniquely suited for wide-window acquisition data. MSFragger-DDA+ provides an efficient and accurate solution for peptide identification, enhancing the detection of low-abundance co-fragmented peptides. Coupled with the FragPipe platform, MSFragger-DDA+ enables more comprehensive and accurate analysis of proteomics data.

Liquid chromatography-mass spectrometry (LC-MS) based proteomics is a widely used, high-throughput method for studying proteins and endogenous peptides. It has been used to study post-translational modifications[1] (PTMs), cancer diseases[2,3], SARS-Cov-2[4], clinical plasma samples[5], protein-protein interactions[6], etc. In the bottom-up proteomics framework, proteins are first digested into shorter peptides to facilitate ionization and fragmentation. These peptides are then separated using liquid chromatography and subsequently analyzed using mass spectrometry. A mass spectrometer typically produces two types of spectra, including mass spectra (MS1) containing ions from intact peptides and tandem mass spectra (MS2)

comprising fragmented ions from selected peptides. The fragmented ions can be used to infer the peptide sequence and PTMs. Two main strategies, data-independent acquisition (DIA) and data-dependent acquisition (DDA), have been developed based on how peptides are selected for fragmentation. In DIA, peptide ions within predefined mass-to-charge (m/z) windows are isolated and fragmented to generate MS2. The isolation windows are designed to cover the entire m/z range within a limited duty cycle. The resulting MS2 spectra are multiplexed, containing fragmented ions from multiple co-eluted peptides. In contrast, DDA isolates selected peptide ions to generate MS2. To ensure high specificity, isolation windows are narrower in DDA

[1]Department of Pathology, University of Michigan, Ann Arbor, MI, USA. [2]Gilbert S. Omenn Department of Computational Medicine and Bioinformatics, University of Michigan, Ann Arbor, MI, USA. ✉e-mail: yufe@umich.edu; nesvi@umich.edu

compared to DIA and typically range from 0.7 to 2.0 Th. However, with complex samples, co-eluting peptide with similar mass may still co-fragment and result in chimeric MS2 spectra[7,8]. Recently, researchers studying single-cell proteomics noted advantages of using a wide-window acquisition[9,10] (WWA) DDA method. The WWA method generates MS2 spectra similar to those in DDA but employs wider isolation windows, allowing for reduced duty cycles. This approach allocates more time for ion accumulation, enabling the production of high-quality tandem mass spectra, particularly beneficial for low-input samples. As a result, more peptides are co-fragmented, producing MS2 spectra as multiplex as those generated by DIA.

Depending on the data acquisition strategy, DDA or DIA, different methods and software have been developed to identify peptides from the acquired MS2 spectra. For DDA, database search-based approaches[11–14] are commonly used to find the best scoring candidate peptide for each MS2. The proteins in the database are first in silico digested into peptides, and then the peptides are in silico fragmented to generate theoretical MS2. The theoretical MS2 are compared with the experimental MS2 to find the most likely peptide-to-spectrum (PSM) match. In doing so, only peptides with the theoretical mass matching (with a narrow tolerance, e.g. 20 ppm) the experimental MS2 precursor peptide masses are considered. The DDA peptide identification methods generally assume that MS2 is not multiplexed, and thus typically only a single peptide identification is reported for each spectrum. However, even with a narrow isolation windows size, there are a substantial number of chimeric MS2 DDA spectra containing co-fragmented peptides[7,8]. Strategies for improved identification of peptides from chimeric spectra have been discussed since the early days of proteomics and include iterative database search (with or without removal of fragments assigned to the top scoring peptide)[15–19] and more elaborated spectral deconvolution[20,21]. Most of these strategies rely on the detection of MS1 precursor peptide features[22–24] of co-fragmented peptides. Although these methods increase the total number of identified peptides from DDA data, they require the co-fragmented peptides to have high-quality precursor peaks to facilitate their identification. Due to different sensitivity of MS1 and MS2, the co-fragmented peptides with high-quality MS2 might have low-quality or even no precursor peaks detected in MS1, hindering their identification.

In contrast, DIA MS2 spectra are assumed to be highly multiplexed, which makes peptide identification more challenging, but spectral library-based[25–27] and library-free[28–30] methods have been developed to tackle the problem. Compared to well-studied peptide identification methods for DDA and DIA data, there are few tools[31] that natively support WWA DDA data. Although WWA DDA MS2 are generated in a similar way to conventional DDA MS2, they are highly multiplexed, making them unsuitable for the existing DDA methods. At the same time, WWA MS2 do not allow extraction of fragment ion chromatograms (XIC), which also makes them unsuitable for most existing DIA software tools.

We have recently described a computational strategy MSFragger-DIA[29] for direct peptide identification from DIA data that essentially blurs the boundary between DDA and DIA data by combining the initial spectrum-centric search of DIA MS2 with a subsequent, peptide-centric re-scoring. Here, we further extend this strategy and present MSFragger-DDA+, a search mode of MSFragger. It utilizes high-resolution DDA MS2 to detect co-fragmented peptides with high sensitivity and accuracy. Unlike other DDA tools, this method does not perform MS1 detection or spectral deconvolution before the database search. Instead, MSFragger-DDA+ performs a full isolation window search and then detects and utilizes MS1 precursor peaks to refine and rescore the results. MSFragger-DDA+ has a lower requirement for the quality of precursor XICs, enabling both high- and low-quality XICs to be detected and utilized for refining PSM scores. We demonstrate that MSFragger-DDA+ is fast, has higher sensitivity than other DDA peptide

identification tools, and fully supports peptide identification from WWA DDA data. We implemented MSFragger-DDA+ as a module in MSFragger, with DDA+ mode automatically triggered by annotating input DDA data as DDA+ type. MSFragger-DDA+ is fully integrated in the widely used FragPipe computational platform to provide a complete solution for proteomics data analysis of DDA and WWA DDA data, from identification to quantification. MSFragger DDA+ has been publicly available as part of MSFragger since version 4.0 (released December 2023), and it has already been used by others in the proteomics community[31,32]. In the following sections, we first provide an overview of the MSFragger-DDA+ algorithm and its integration into the FragPipe computational platform. Next, we evaluate its performance across various datasets, comparing its sensitivity, speed, and accuracy with existing methods. We conclude by discussing the implications of the MSFragger-DDA+ approach and related topics. Finally, the MSFragger-DDA+ algorithm and the experiments' parameters are detailed in the Methods section.

## Results

### Overview of MSFragger-DDA+

In the database search framework, traditional peptide identification algorithms search each MS2 against candidate peptides within a narrow mass tolerance, which is normally 5–20 ppm (for high resolution MS2 data) around the precursor mass reported by a mass spectrometer for that scan, to identify the best match. In contrast, MSFragger-DDA+ searches each MS2 against all peptides within the full isolation window (Fig. 1). During the database search, MSFragger-DDA+ uses hyperscore[11,33] to measure the similarity between the MS2 and the candidate peptides in the isolation window. It is reasonable to assume that there will be many false matches since MSFragger-DDA+ does not constrain the search within the narrow mass tolerance of the peptide selected by the mass spectrometer for MS2. At the same time, many of the co-fragmented peptides do not have a high-quality precursor peak observed in the parent MS1 scan. Thus, we developed post-database search refinement steps, including precursor XIC detection, shared fragment removal, and rescoring. MSFragger-DDA+ detects and extracts XICs for each PSM after the database search. It extracts as many isotopic XICs as possible to compare the intensity distribution with the theoretical distribution[34] with the Kullback–Leibler divergence[35]. Because the theoretical m/z and charge are known after the database search, MSFragger-DDA+ extracts the XICs in the "targeted" manner. In contrast to the "untargeted" MS feature detection approach[22,28,36] that may struggle to accurately detect and extract low-abundance features and their peak curves, the targeted approach reliably extracts the XICs given the theoretical m/z values. This extraction is restricted to the retention time range associated with the corresponding MS2 scan, ensuring high specificity and accuracy. Moreover, the targeted approach does not suffer from the challenges of untargeted deconvolution of overlapped isotopic peak clusters which might result in incorrect charge and m/z determination. Thus, the targeted XIC detection is more sensitive, even when precursor XICs are of low quality. It maximizes the potential to identify co-fragmented peptides with low-abundance precursor signals. After the targeted extraction, PSMs with low-quality XICs are discarded. Then, MSFragger-DDA+ rescores the PSMs by removing the fragments shared by multiple peptides using a greedy algorithm[29]. Given a list of PSMs derived from the same tandem mass spectrum, the greedy algorithm operates iteratively. First, it removes the fragment peaks matched to the top-scoring PSM. Next, it eliminates the top-scoring PSM from the list, re-calculates the hyperscores for the remaining PSMs, and re-ranks them accordingly. These steps are repeated until there are insufficient fragment peaks to match any PSM. Upon completing all data analysis steps, it generates output files that can be processed by downstream tools. MSFragger-DDA+ has been highly optimized for fast processing speeds and efficient memory usage. For most datasets, it can complete

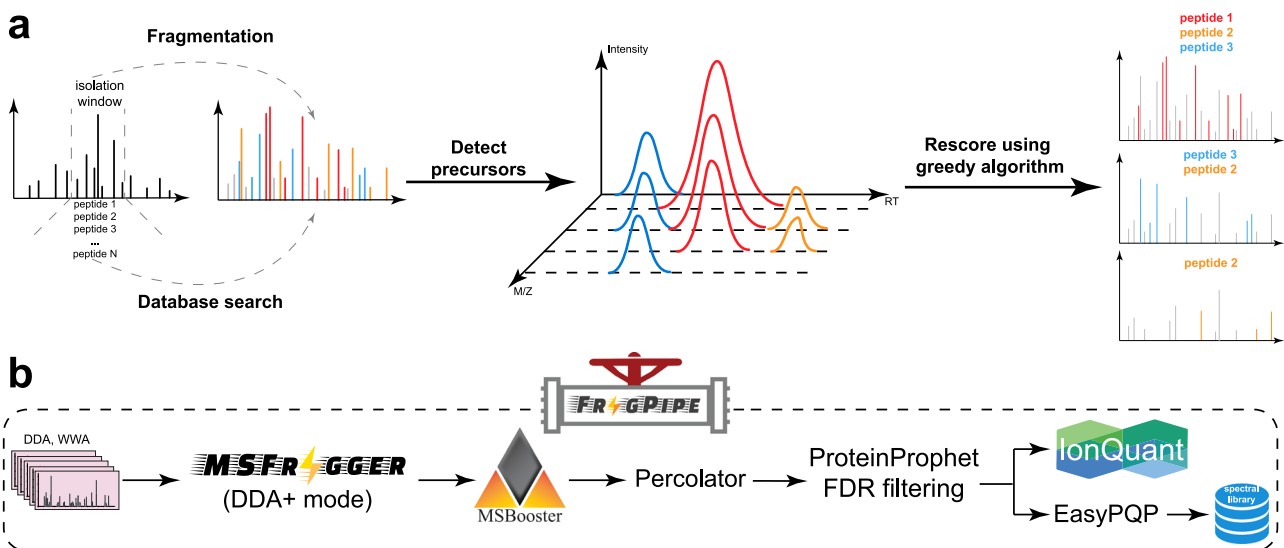

**Fig. 1 | Overview of MSFragger-DDA+ and FragPipe. a** MSFragger-DDA+ algorithm. Each tandem mass spectrum is searched against all peptides within the isolation window. Subsequently, MSFragger-DDA+ detects the precursor signals for each of the matched PSMs. After filtering out the PSMs with low-quality precursor signals, MSFragger-DDA+ rescores the PSMs using a greedy algorithm.

**b** MSFragger-DDA+ workflow in FragPipe software. The workflow contains MSFragger-DDA+ for database searching, MSBooster for deep-learning-based rescoring, Percolator for PSM ranking, PorteinProphet for protein inference, FDR filtering, IonQuant for label-free quantification (optional), and EasyPQP for spectral library generation (optional). Source data are provided as a **Source Data** file.

a tryptic database search on a standard desktop computer equipped with 32 GB of memory and a mainstream CPU. For tasks such as phosphopeptide or non-specific digested peptide identification, a larger memory size, such as 64–128 GB, is recommended.

To make MSFragger-DDA+ easy to access and user-friendly, we have integrated it into FragPipe computational suite. The output can be seamlessly processed by FragPipe modules, including MSBooster[37] deep-learning-based rescoring, Percolator[38] PSM re-ranking, PeptideProphet[39] PSM rescoring, PTMProphet[40] modification localization, ProteinProphet[41] protein grouping, Philosopher[42] false discovery rate (FDR) filtering, IonQuant[35] quantification, EasyPQP spectral library generation, PDV[43] and Skyline[25] visualization. MSFragger-DIA+ mode can be triggered in any FragPipe workflow that supports DDA by simply annotating the input MS files as DDA+ type.

## MSFragger-DDA+ improves the sensitivity of peptide identification from DDA data

First, we evaluated the performance of MSFragger-DDA+ using two DDA datasets. The first one was published by Searle et al.[44]. There are five samples with different normalized collision energy (NCE): 22, 27, 32, 37, and 42. We used MaxQuant[36] (version 2.4.13), MetaMorpheus[24] (version 1.0.5), MSFragger[11] (version 4.1) in DDA mode, and MSFragger-DDA+ (version 4.1) for peptide identification. We also included Scribe's results from the original publication[44]. The second DDA dataset was published by Richards et al.[45]. There are six samples with different types of enzymatic digestions (trypsin, AspN, and GluC) and fragmentations (CID and HCD). For MetaMorpheus, the precursor deconvolution was enabled by default to deconvolute co-fragment peptides separately. For MSFragger and MSFragger-DDA+, FragPipe (version 22.0) was used to perform MSBooster deep-learning-based rescoring, Percolator re-ranking, ProteinProphet protein grouping, and Philosopher FDR filtering (see "Methods" for detail).

The numbers of identified peptide sequences after filtering at a 1% false discovery rate (FDR) for the Searle et al. and Richards et al. datasets are shown in Fig. 2a, b, respectively. We used the FDR reported by the tools to perform the filtering ("Methods"). Figure 2a shows that although MetaMorpheus, MSFragger (conventional DDA mode), and Scribe have similar performance, the sensitivity of MSFragger-DDA+ is much higher across all NCE. On average,

MSFragger-DDA+ identified 57% more peptide sequences than MSFragger in the conventional DDA mode. Figure 2b shows similar comparisons, with MSFragger-DDA+ identifying from 15% (AspN CID and GluC CID) to 45% (Trypsin HCD) more peptide sequences depending on the combination of enzymatic digestion and fragmentation. We also compared the runtime of MaxQuant, MetaMorpheus, MSFragger in DDA mode, and MSFragger-DDA+ (Fig. 2c). The comparison shows that MaxQuant and MetaMorpheus have similar speed, whereas MSFragger-DDA+ coupled with FragPipe is four times faster. Because during the revision of this manuscript MetaMorpheus released a new version 1.0.6, we re-analyzed the dataset using the new version and observed similar results (Supplementary Fig. 2a).

## MSFragger-DDA+ coupled with FragPipe has good FDR control

The previous section demonstrated that MSFragger-DDA+ identifies many more peptide sequences compared to MSFragger in the conventional DDA mode and the other tools tested. We performed an entrapment database search[46] to evaluate the FDR control. The entrapment database contains the entrapment sequences that do not exist in the original database. Given the original target database, we generated an entrapment sequence for each target protein sequence by random shuffling. During the shuffling, the peptide C-termini were fixed. The target and entrapment sequences were used as the new target database. We searched the Searle et al.[44] dataset against the new database using MaxQuant, MetaMorpheus, MSFragger in DDA mode, and MSFragger-DDA+. For MSFragger and MSFragger-DDA+, FragPipe was used to perform downstream analysis as described in the previous section (see also "Methods"). Figure 2d shows the number of proteins, entrapment proteins, and two false discovery proportion (FDP) estimations proposed by Wen et al.[46]. To make it more intuitive, we used the name "upper bound" to replace the name "combined method" used by Wen et al.[46] since it is a method to calculate the upper bound of the FDP. The peptide level statistics are shown in Supplementary Fig. 1. This experiment demonstrates that not only MSFragger-DDA+ identifies the largest number of target peptide sequences and proteins it also has a similarly low FDP as MSFragger and lower FDP than that for the other tools. We also checked that using the latest MetaMorpheus 1.0.6 resulted in minor changes only (Supplementary Fig. 2b).

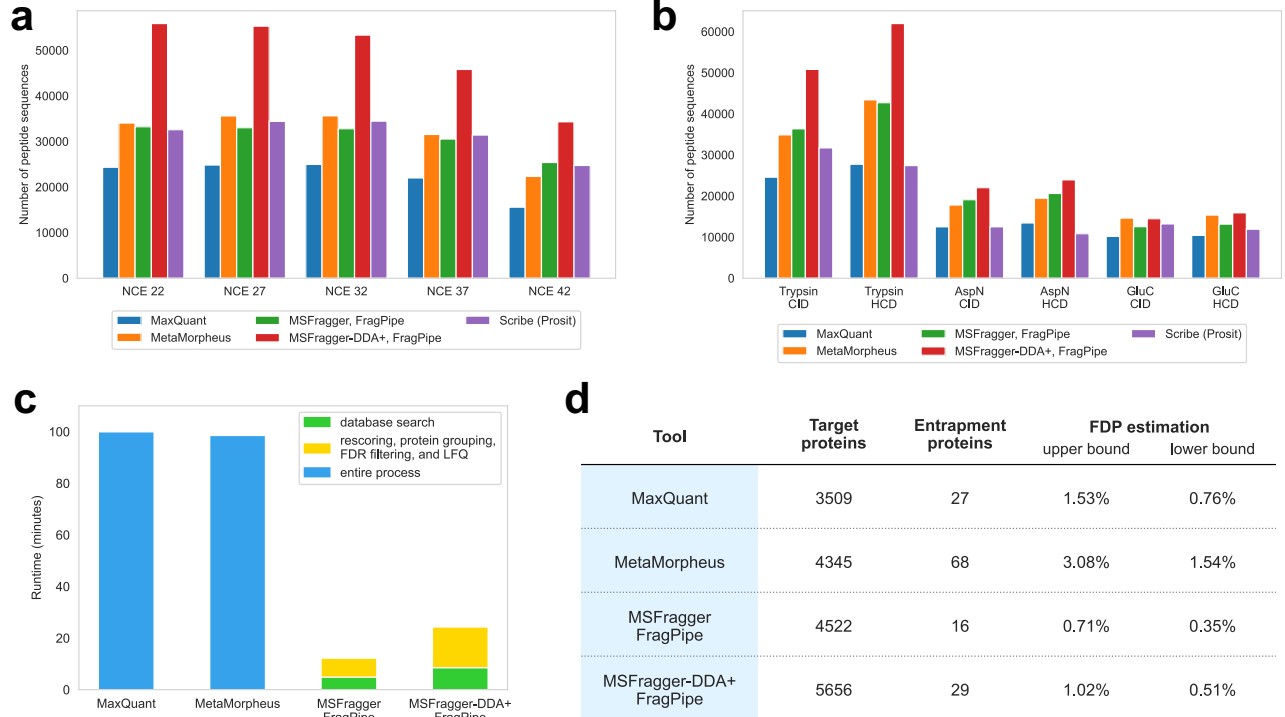

**Fig. 2 | Sensitivity, speed, and false discovery proportion assessment using traditional DDA datasets with different NCE and enzymatic digestions.**
**a**, **b** Number of peptide sequences identified by MaxQuant, MetaMorpheus, MSFragger, MSFragger-DDA+, and Scribe. **a** Data with different NCE. **b** Data with different enzymatic digestions. **c** Runtime of MaxQuant, MetaMorpheus,

MSFragger, and MSFragger-DDA+. **d** Protein-level FDP evaluation for MaxQuant, MetaMorpheus, MSFragger, and MSFragger-DDA+. Two calculation methods, including the upper bound and lower bound, were applied. Source data are provided as a **Source Data** file.

## Application to DDA-PASEF data

We used a dataset published by Wang et al.[47] to demonstrate that MSFragger-DDA+ performs well when analyzing DDA-PASEF data generated using the Bruker's timsTOF platform. There are two cell lines, A549 and K562. Each cell line has three biological replicates, and each biological replicate has four technical replicates. We used MSFragger in DDA mode and MSFragger-DDA+ to perform the peptide identification. FragPipe was used to process the peptide identification outputs as in the previous sections. For label-free quantification (LFQ), IonQuant[35,48] with and without match-between-runs (MBR) was used. Figure 3a and Supplementary Fig. 3a show the number of quantified proteins from the two cell lines, respectively. A protein is quantified when it has a non-zero intensity. For each cell line, three biological replicates with and without MBR, are listed separately. For each biological replicate, different color transparency levels indicate the number of proteins quantified in one to four technical replicates. These two figures show that MSFragger-DDA+ coupled with FragPipe has higher sensitivity than the DDA workflow. Without MBR, the DDA+ workflow quantified 16–19% more proteins among the six experimental replicates. The differences are smaller with MBR because it transfers identifications from one replicate to another to reduce missing values, however, the DDA+ workflow still quantified 11–13% more proteins.

The quality of quantification was also evaluated by calculating the coefficient of variations (CVs) using the four technical replicates of each sample (see Fig. 3b and Supplementary Fig. 3b). Overall, the proteins identified uniquely with one method (MSFragger-DDA+ or MSFragger) have higher CVs than the proteins identified in common. The same set of common proteins quantified by both DDA and DDA+ workflows do not have significantly different CVs. The proteins unique to the DDA+ workflow have slightly higher CVs than the proteins unique to DDA. We reasoned that these proteins are of lower

abundance and have low signal-to-noise ratios because their peptides were co-fragmented but not dominant in the MS2. There are also more DDA+ unique proteins than the DDA unique proteins. Figure 3c and d show the proportion of missing values in the DDA and DDA+ workflows, respectively. Figure 3c is without MBR, and Fig. 3d is with MBR. The results show that the DDA+ workflow quantified notably more proteins (6585 vs 5619 without MBR, and 6725 vs 5916 with MBR) and with a comparable missing value proportion (17% without MBR and 7% with MBR) as the conventional MSFragger DDA workflow. Supplementary Fig. 3c, d show the principal component analysis (PCA) plots from the DDA and DDA+ workflows, respectively, exhibiting comparable separation between the samples (and similar to that reported in the original publication).

## MSFragger-DDA+ fully supports wide-window acquisition data

WWA[9,10] is a data acquisition approach primarily designed for single-cell proteomics. It generates MS2 in the DDA mode, but with wider isolation windows. Multiple peptides are intentionally co-fragmented to produce multiplexed spectra. As a result, traditional DDA database search tools are not well-suited for such data. Using the datasets published by Truong et al.[9] and Matzinger et al.[10], we demonstrate that MSFragger-DDA+ has excellent performance on such data.

The first dataset[9] contains 29 low-input samples with different isolation window sizes (1.6, 2, 4, 8, 12, 18, 24, and 48 Th), maximum injection time (54, 86, 118, and 246 ms), and MS2 resolutions (30 K, 45 K, 60 K, and 120 K). Each sample has two technical replicates. We used CHIMERYS[31] coupled with Proteome Discoverer, MetaMorpheus[24], and MSFragger-DDA+ for peptide identification. CHIMERYS natively supports the WWA data type. For MetaMorpheus, the precursor deconvolution was enabled to deconvolute co-fragmented peptides into separated spectra. Figure 4a shows the number of peptides identified from the samples with 54 ms

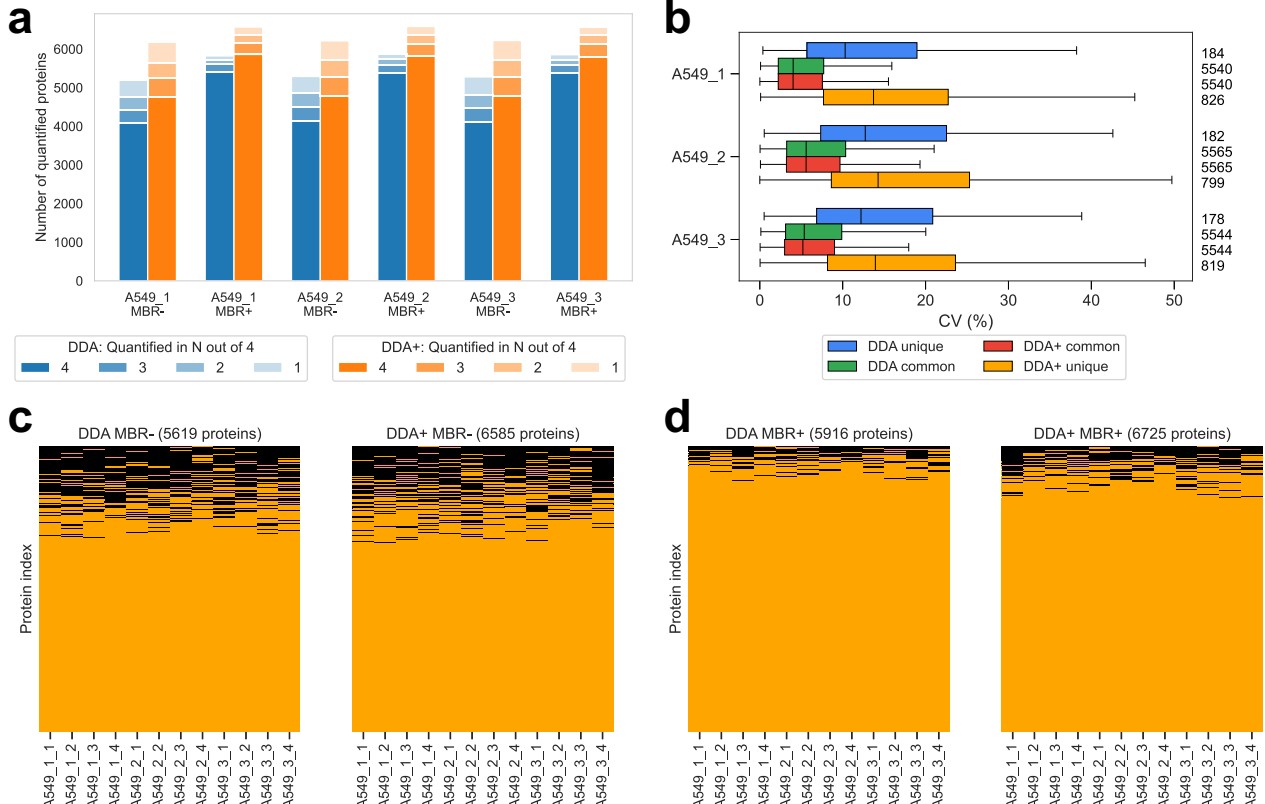

**Fig. 3 | Performance benchmarking using timsTOF ddaPASEF data. a** Number of quantified proteins from the DDA and DDA+ workflows in the A549 cell line dataset (with three biological replicates, and four technical replicates for each). "MBR+" and "MBR-" refer to IonQuant run with and without MBR, respectively. **b** Numbers and CVs of overlapped and non-overlapped proteins quantified from the DDA and DDA+ workflows using the A549 cell line. The blue box plots are from the unique proteins of the DDA mode, the green box plots are from the common proteins of the DDA mode, the red box plots are from the common proteins of the DDA+ mode, and the yellow box plots are from the unique proteins of the DDA+ mode. The common proteins are the overlapping proteins quantified in both DDA and DDA+ modes. The numbers on the right are the quantified proteins. For biological replicates A549_1, A549_2, and A549_3, there are 184, 182 and 178 unique proteins of

the DDA mode; 826, 799, and 819 unique proteins of the DDA+ mode; and 5540, 5565, and 5544 common proteins, respectively. The box in each plot captures the interquartile range (IQR) with the bottom and top edges representing the first (Q1) and third quartiles (Q3), respectively. The median (Q2) is indicated by a horizontal line within the box. The whiskers extend to the minima and maxima within 1.5 times the IQR below Q1 or above Q3. **c** Two plots showing the protein non-missing values and missing values from the DDA and DDA+ workflows. A549 cell line data. MBR is disabled. The number of columns equals to the number of proteins quantified in the specific setting and listed at the top of each plot. The proteins with non-zero intensities are in orange, and the proteins with zero intensities are in black. **d** Similar to (**c**) but the MBR is enabled. Source data are provided as a **Source Data** file.

maximum injection time, 30 K MS2 resolution, and 2 - 48 Th isolation window sizes. Figure 4b shows the number of identified peptides from the samples with 118 ms maximum injection time, 60 K MS2 resolution, and 2 - 48 Th isolation window sizes. To avoid redundancy, the figures for other samples are not shown, but they demonstrate a similar trend. The result files can be found as described in "Data availability" section. CHIMERYS and MSFragger-DDA+ obtained similar numbers, and higher than that of MetaMorpheus. Since this dataset has another four high-input samples to be used as "library" runs in the MBR, we performed the same analysis with MBR enabled (Supplementary Fig. 4a, b, and Supplementary Fig. 5a–d for comparison with MetaMorpheus version 1.0.6). For MSFragger-DDA+, the MBR was performed using IonQuant as part of FragPipe. For CHIMERYS and MetaMorpheus, the MBR was performed by Proteome Discoverer and MetaMorpheus, respectively. The figures show that MetaMorpheus reported the highest number of peptides after applying MBR, followed by MSFragger-DDA+ coupled with FragPipe. CHIMERYS coupled with Proteome Discoverer reported the lowest number compared to the other two. However, considering that different tools have different parameters and criteria to perform and quality control the MBR results, it is difficult to make a fair comparison when MBR is enabled.

The second dataset is also from Truong et al.[9]. There are two cell lines: K562 and HeLa. Each cell line has two biological replicates with different gradient lengths of 20 min and 40 min. Each biological replicate has eight technical replicates. We used CHIMERYS coupled with Proteome Discoverer 3.0, MetaMorpheus, and MSFragger-DDA+ coupled with FragPipe to analyze the data. As there are eight technical replicates, we evaluated the quantification quality by calculating the CVs for each biological replicate. Figure 4c and d show the number of quantified proteins. Similar to the first dataset, there are four high-input samples to be used as "library". Supplementary Fig. 4c, d show bar plots with the MBR enabled. CHIMERYS coupled with Proteome Discoverer and MSFragger-DDA+ coupled with FragPipe have similar sensitivity when the MBR is not enabled. Both have higher sensitivity than MetaMorpheus. When the MBR is enabled, MSFragger-DDA+ coupled with FragPipe and MetaMorpheus have similar sensitivity, which is higher than that of CHIMERYS coupled with Proteome Discoverer. The comparisons were not affected by switching to the latest release of MetaMorpheus version 1.0.6 (Supplementary Fig. 5e–h).

The third dataset is from Matzinger et al.[10]. There are 43 samples with different isolation window sizes (1, 2, 3, 4, 5, 6, 7, 8, 12, 18, 24, 28, 56 Th) and sample amounts (250 pg, 1 ng, 10 ng, 200 ng, and 400 ng). Each sample has three technical replicates. We

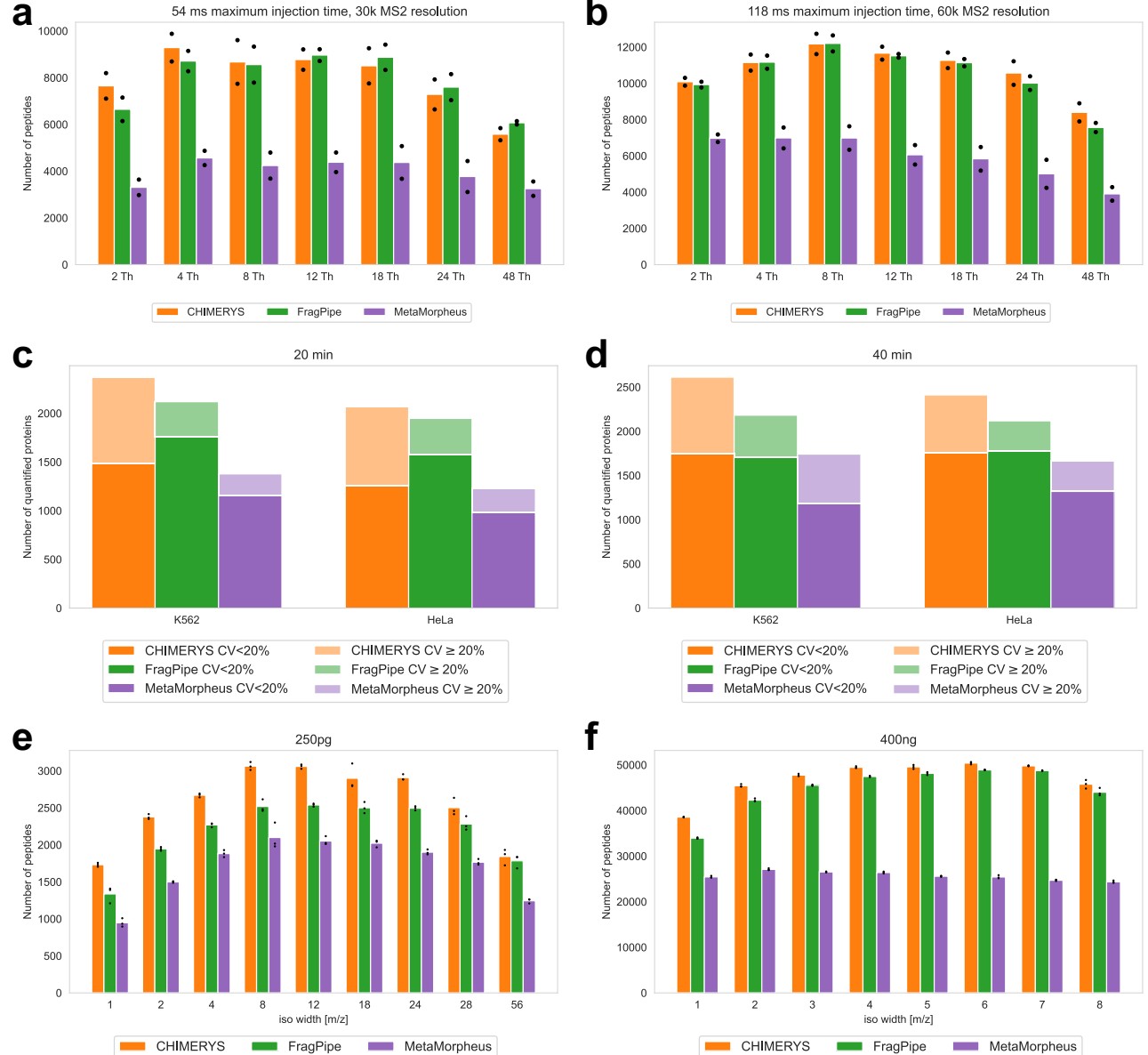

**Fig. 4 | Sensitivity assessment using three WWA datasets. a, b** Numbers of peptides from the first WWA dataset of Truong et al. There are 14 samples with different isolation windows, maximum injection time, and MS2 resolutions. Each sample contain two technical replicates. MBR is disabled. The bar height represents the mean of the counts, and the black dot represents the peptide count for each replicate. **c, d** Numbers and CVs of quantified proteins from the second dataset of Truong et al. The dark color is with CV < 20% and the light color is with CV ≥ 20%.

The samples are from K562 and HeLa cell lines, respectively. There are four samples with different combinations of cell line and gradient length. Each sample has eight technical replicates. MBR is disabled. **e, f** Numbers of identified peptides from the WWA dataset published by Matzinger et al. There are 17 samples with different isolation windows and sample amounts. Each sample has three technical replicates. The bar height represents the mean of the counts, and the black dot represents the peptide count for each replicate. Source data are provided as a **Source Data** file.

used MSFragger-DDA+ coupled with FragPipe, and MetaMorpheus to analyze the dataset. We also used the CHIMERYS/Proteome Discoverer 3.0 results as provided by the authors as part of the original publication. Figure 4e, f, Supplementary Fig. 4e, f show the number of identified peptides from samples with 250 pg, 1 ng, 200 ng, and 400 ng. The samples with 10 ng are not shown due to redundancy. The result files can be found as described in "Data availability" section. CHIMERYS coupled with Proteome Discoverer and MSFragger-DDA+ coupled with FragPipe have similar sensitivity, and higher than that of MetaMorpheus. From the samples with smallest amount (250 pg), CHIMERYS, MSFragger-DDA+, and MetaMorpheus identified on average 2562, 2186, and 1713 peptides, respectively. From the samples with the largest amount (400 ng), CHIMERYS, MSFragger-

DDA+, and MetaMorpheus identified on average 47100, 44879, and 25689 peptides, respectively.

The above datasets from two laboratories with different sample preparation and data acquisition configurations demonstrate that MSFragger-DDA+ performs well for identifying peptides from WWA data. Furthermore, MSFragger-DDA+ is faster than other tools when analyzing WWA dataset. For the first dataset with 62 mzML files as input, MSFragger-DDA+ coupled with FragPipe took 68.8 min on a Linux server with Intel Xeon Gold 6354 CPU (3.00 GHz, 36 physical cores) and 768 GB RAM (although 768 GB RAM was available, only a small proportion was used during the analysis). In contrast, CHIMERYS and Proteome Discoverer took 25 h in total, of which most of the time (16.5 h) was taken by CHIMERYS running on the company's proprietary

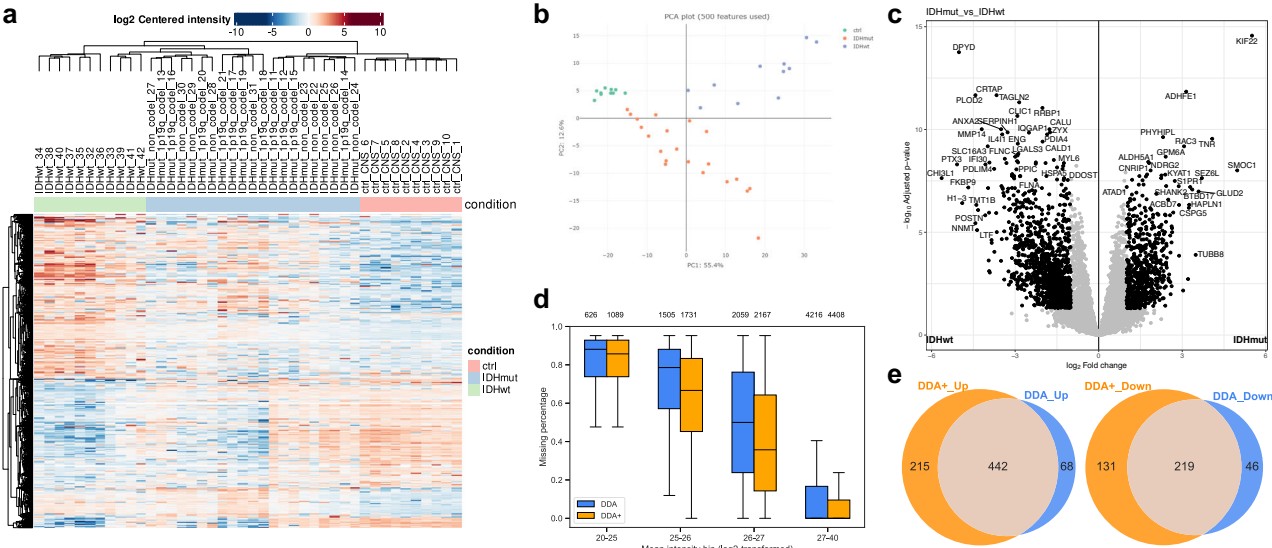

**Fig. 5 | Performance demonstration using the glioma dataset. a** Heatmap of the gene intensities from the DDA+ workflow. **b** PCA plot of the quantitative results from the DDA+ workflow. **c** Volcano plot comparing 10711 gene intensities between IDHmut samples (*n* = 21) and IDHwt samples (*n* = 11) from the DDA+ workflow. Log2 fold change was calculated using limma's moderated *t* test. The *p* values were adjusted using the Benjamini-Hochberg procedure. The black dot represents the differentially expressed genes based on a fold change threshold of 2 and adjusted *p* value of 0.05. **d** Box plots showing the percentage of protein-level missing values from the DDA and DDA+ workflows. A total of 9395 proteins from DDA+ workflow

and 8406 proteins from DDA workflow are compared across the defined intensity ranges, with the protein count for each box displayed at the top. The box in each plot captures the IQR with the bottom and top edges representing the Q1 and Q3, respectively. The median (Q2) is indicated by a horizontal line within the box. The whiskers extend to the minima and maxima within 1.5 times the IQR below Q1 or above Q3. **e** Venn diagrams showing the number of upregulated and down-regulated genes in the DDA and DDA+ workflows. Source data are provided as a **Source Data** file.

cloud computing platform. Although a fair runtime comparison is difficult, our experiment showed that MSFragger-DDA+ coupled with FragPipe was at least 20 times faster than CHIMERYS coupled with Proteome Discoverer in these data.

## MSFragger-DDA+ has higher sensitivity in analyzing phosphoproteome data

To demonstrate the enhanced capability of MSFragger-DDA+ in identifying peptides from phosphorylation-enriched data, we benchmarked MSFragger-DDA+ against MSFragger in DDA mode using the dataset published by Sacco et al.[49] ("Methods"). The dataset includes six replicates of a human sample. MSFragger-DDA+ identified 22378 phosphosites, compared to 21,249 identified by MSFragger in DDA mode, with an overlap of 20018 phosphosites between the two modes. After stripping modifications, MSFragger-DDA+ identified 23,948 phosphosequences, whereas MSFragger in DDA mode identified 21,070, with 19,907 overlapping phosphosequences. These results demonstrate that MSFragger-DDA+ identified ~5% more phosphosites and ~14% more phosphosequences than the traditional DDA mode. The improvement is less pronounced compared to the results from proteome data. This difference is likely due to the larger search space associated with phosphoproteome peptide identification, which reduces the sensitivity under the same FDR control. Additionally, we applied a 0.75 localization probability cut-off for phosphosites. Since co-fragmented peptides normally have lower fragmentation quality and localization confidence, this cut-off further narrowed the difference between the DDA+ and DDA modes.

## MSFragger-DDA+ leads to detection of more differentially expressed proteins and genes

We further demonstrate the performance of MSFragger-DDA+ combined with the FragPipe suite by utilizing a dataset from IDH-mutated glioma patients[50]. The dataset includes three groups of samples: IDH-mutant group (11 1p/19q-codeleted samples and 10 astrocytoma

samples), one IDH wild-type group (11 samples), and one non-neoplastic CNS tissue control group (10 samples). Further details about sample preparation and data acquisition can be found in the original publication. We utilized MSFragger-DDA+ for peptide identification as part of the FragPipe's LFQ-MBR built-in workflow. Following the database search, FragPipe performed the PSM rescoring[37], protein grouping[41], FDR filtering[42], and LFQ[35] ("Methods"). FragPipe with MSFragger-DDA+ identified 10738 proteins, 935 more proteins than MSFragger (and 1231 more proteins than MaxQuant reported in the original publication). Proteins with non-zero MaxLFQ intensities were used to perform downstream analysis using FragPipe-Analyst[51]—a recent addition to the FragPipe family of tools that enables seamless analysis of FragPipe generated quantification data. Among the FragPipe-Analyst's output files, the heatmap (Fig. 5a) successfully recovered distinct protein expression patterns for the three sample groups (IDH wild-type, IDH mutant, and control). The PCA (Fig. 5b) plot demonstrated that the IDH wild-type gliomas were most distinct from the control group and were separated from the IDH mutant cases, as expected. We then used FragPipe-Analyst to generate a volcano plot (Fig. 5c) of the IDH mutant versus wild-type. After filtering the genes with 0.05 Benjamini-Hochberg adjusted *p*-value and 1 log2 fold change, the differentially expressed genes highly overlapped with those reported in the original publication. We also compared the results of the DDA and DDA+ workflows. The DDA+ workflow quantified more proteins (counting unique gene symbols) (Supplementary Fig. 6a) and with fewer missing values (Fig. 5d and Supplementary Fig. 6b), leading to the detection of more differentially expressed proteins after Limma analysis (Fig. 5e). Finally, we took advantage of the Enrichr[52] enrichment analysis tool integrated into FragPipe-Analyst. We performed Gene Ontology (GO) analysis using proteins differentially expressed between the IDH wild-type and the mutant samples (Supplementary Fig. 6c). Three levels of GO analysis show that the DDA+ workflow identified more differential proteins corresponding to each of the enriched categories. Overall, our analysis suggests that MSFragger-

DDA+ coupled with the FragPipe leads to more sensitive detection of differentially expressed proteins and pathways in cancer proteomics profiling experiments.

## Discussion

We presented a peptide identification method, MSFragger-DDA+, that has a higher sensitivity compared to the conventional DDA peptide identification strategy. Unlike conventional tools developed for DDA data, it starts by searching MS2 spectra against all peptides within the full isolation window to enable identification of all possible co-fragmented ions. Importantly, many of those co-fragmented peptides do not have a strong precursor MS1 signal, making them difficult to identify from chimeric DDA spectra using strategies that rely on the knowledge of the masses of co-fragmented peptides prior to the search. Instead, in MSFragger-DDA+, precursor signals are considered only at the second, targeted rescoring step. At that stage, a small list of the top scoring candidate peptides for each MS2 spectrum has been established, and the XICs for these peptide ions can be more easily extracted from MS1 data in a targeted way. Thus, by reversing the order of the two key steps, database search and MS1 feature detection, MSFragger-DDA+ can detect more of low abundant, co-fragmented peptides.

The application of MSFragger-DDA+ to WWA data has significant implications in single-cell proteomics. Single-cell proteomic studies often require the analysis of low-input samples, whereas traditional DDA approaches may struggle with peptide identification because of limited ion signal strength and suboptimal fragmentation quality. By utilizing an enlarged isolation window size, the duty cycle is reduced, allowing more time for ion accumulation. This ensures sufficient ions to generate abundant fragmented peaks, resulting in high-quality yet chimeric tandem mass spectra. MSFragger-DDA+'s capability to sensitively identify co-fragmented peptides from these chimeric spectra significantly expands the depth of proteome coverage achievable in single-cell experiments. Moreover, MSFragger-DDA+ is substantially faster than the existing tools that support WWA data, an advantage that is particularly crucial for single-cell studies involving large datasets with extensive analysis requirements. This speed enhancement addresses a critical bottleneck in data processing, thereby enabling more practical and high-throughput single-cell proteomics workflows that use WWA method.

MSFragger-DDA+ is the latest tool in our long-going efforts to develop a computational peptide identification workflow that provides a uniform treatment of MS2 data across the entire spectrum of data acquisition strategies, from conventional DDA to narrow-window DIA, to wide-window DDA, and wide-window DIA. The key to these efforts has been our development of the fragment ion indexing algorithm that enabled spectrum-centric search of MS2 data against protein sequence databases in essentially unrestricted way. In the original MSFragger[11] manuscript we demonstrated the application of fragment ion indexing to open (also known as mass tolerant or unrestricted) searches of DDA data to identify modified peptides. Later we have extended this work in MSFragger-LOS[53] for localization-aware open search, in MSFragger-Glyco[54] for glycopeptide identification, and in MSFragger-Labile[55] for labile modifications. In parallel, we have applied our strategy to enable direct, spectrum-centric search of DIA data with MSFragger-DIA[29]. With MSFragger-DDA+, we now return to the analysis of conventional DDA data and enable the identification of co-fragmented peptides from chimeric DDA spectra using full isolation window search. Regardless of the MS2 data type, we start with the spectrum-centric search of MS2 spectra against the theoretical spectra generated from the protein sequence database, without the need for a spectral library or any predictions of peptide properties or MS2 spectra prior to the search. It is only after the MSFragger database search that we leverage deep learning-based fragment intensity and retention time predictions (and only when such predictions are likely

to be accurate) in MSBooster to refine and rescore the peptide candidates to boost the identification sensitivity[37]. Furthermore, regardless of the MS2 data type, all MSFragger search results are processed using the same downstream tools (MSBooster with Percolator for rescoring; protein inference with ProteinProphet; FDR filtering with Philosopher). To summarize, with the spectrum-centric database search framework at its core, the MSFragger family of tools unifies peptide identification across different MS2 data acquisition modalities (DDA, wide window DDA, DIA) and different search modes (closed, open, and mass-offset).

There are still important differences between the full isolation window search described in this work and the open search for modified peptides using DDA data. Given an MS2, MSFragger-LOS performs an open search of conventional DDA spectra using a very wide mass window (e.g. −150 to 500 Da, much larger than the isolation window) around the precursor mass associated with the corresponding DDA MS2 scan to identify modified (mass shifted) peptides. In doing so, it matches the fragment peaks shifted by unknown modifications, in addition to matching the unshifted peaks. The open modification search normally reports one peptide for each MS2 and does not consider co-fragmented peptides. On the contrary, MSFragger-DDA+ searches for peptides that fall within the isolation window, without allowing any unexpected mass shifts (larger than the isolation window width) or using shifted peaks in scoring. MSFragger-DDA+ then detects the peptide precursor signals after searching the database to refine the matches and report the observed precursor m/z and charge that may be different from that listed for the DDA spectrum. Enabling the full isolation window search for co-fragmented peptides in parallel with the open search for modified peptides is the next computational challenge that we plan to address in future work.

With the development of MSFragger-DDA+, all workflows developed in FragPipe for the analysis of DDA data can now be run using DDA+ mode. In our experience, MSFragger-DDA+ provides a higher sensitivity boost with unfractionated single-shot LC-MS runs compared to fractionated ones. This is expected given that an unfractionated run, keeping the LC gradient constant, contains more co-eluting peptides, and thus more chimeric spectra, than an MS run on a fractionated peptide sample. Furthermore, our experiments have shown that MSFragger-DDA+ significantly reduces the number of missing intensity values across all samples in multi-sample analyses typical to label-free quantification workflows. However, when LFQ analysis is performed with MBR enabled, the differences became smaller because MBR helps to achieve more complete quantification matrix via the transfer of peptide identifications between the runs. Nevertheless, MSFragger-DDA+ still results in more peptides and proteins in total, which results in better downstream analysis results. Finally, not all DDA-based workflow, most notably quantitative workflows based on isobaric labeling such as tandem mass tag (TMT), would benefit from an increase in the number of peptide identifications afforded by MSFragger-DDA+. In TMT workflows, co-fragmented peptides can introduce interference in the isobaric quantification because they share the same set of reporter ions. To address this issue, most computational workflows for TMT data (including TTM-Integrator in FragPipe) by default discard MS2 spectra with low isolation purity scores[56]. Therefore, unlike LFQ, it may not be beneficial to detect co-fragmented peptides using the MSFragger-DDA+ approach in TMT-based quantitative proteomics datasets. In scenarios involving large search spaces, such as non-specific searches for human leukocyte antigen peptides, the full isolation window approach may present computational challenges and result in reduced sensitivity owing to the extensive number of candidate peptides. Furthermore, the current version of MSFragger-DDA+ does not support open or mass-offset searches, thereby limiting its applicability in the identification and discovery of unknown PTMs.

## Methods

### DDA+ enhanced peptide identification

MSFragger-DDA+ supports both DDA and WWA data types. In contrast to the traditional database search approach that matches a spectrum against peptides within a narrow mass window, MSFragger-DDA+ matches all peptides in the isolation window to detect all possible co-fragmented peptides. Hyperscore[11,33] is used to measure the peptide-spectrum similarity during this process. After the database search, MSFragger-DDA+ detects and extracts the precursor XICs for each matched peptide. Fragment indexing[48] is used to accelerate this procedure. MSFragger-DDA+ extracts as many isotopic XICs as possible, and compares the intensity distribution with theoretical distribution[34] using Kullback-Leibler divergence[35]. Peptides with low-quality XICs are discarded. Last, all peptides are rescored using the greedy algorithm used by Yu et al.[29].

### Traditional DDA data

Two DDA datasets published by Searle et al.[44] and Richards et al.[45] were used to evaluate the performance of MSFragger-DDA+. The first dataset was generated by a Thermo Exploris 480 mass spectrometer. There are five HeLa samples with different NCEs: 22, 27, 32, 37, and 42. The second dataset was generated by a Thermo Orbitrap Fusion Lumos mass spectrometer. There are six HEK 293 T samples with different enzymatic digestions (trypsin, AspN, and GluC) and fragmentations (CID and HCD). Details of sample preparation and data acquisition can be found in the original publications. The data was analyzed using MaxQuant[36] (version 2.4.13), MetaMorpheus[24] (version 1.0.5), Scribe[44], MSFragger[11] (version 4.1) in DDA mode, and MSFragger-DDA+ (version 4.1). The FASTA database is the Homo Sapiens reference proteome used by Searle et al.[44] (20361 proteins). MaxQuant was run using default settings without including built-in contaminants. The "evidence.txt" file with decoys removed was used to count the peptide sequences. For MetaMorpheus, the precursor deconvolution was enabled by default to deconvolute co-fragmented peptides into separated PSMs. We used the "*_Peptides.psmtsv" files in the "Individual File Results" folder to count the peptide sequences identified from each input file. The decoys were removed, and the peptides were filtered with "PEP_QValue < 0.01". For MSFragger and MSFragger-DDA+, the reversed decoy sequences were generated and appended to the target database. FragPipe (version 22.0) was used to process the outputs of MSFragger and MSFragger-DDA+ to perform MSBooster deep-learning-based rescoring, Percolator PSM re-ranking, ProteinProphet protein grouping, and Philosopher FDR filtering. The "Default" workflow was applied with adjusted maximum allowed missed cleavage and enzymatic rules. We used the "peptide.tsv" files, which were filtered at 1% peptide- and protein-level FDR, to count the peptide sequences. For all tools, the maximum allowed missed cleavage was set to 1. Acetylation of the protein N-terminus and oxidation of methionine were set as variable modifications, and carboxymethylation of cysteine was set as a fixed modification. The results of Scribe were downloaded from Searle et al.[44]. The parameter, log, and result files are available as described in "Data availability" section. The scripts to summarize the results and generate the figures are available as described in "Code availability".

### False-discovery rate evaluation using entrapment database search

An entrapment database was used to evaluate the FDR of MSFragger-DDA+ and compare with that from the existing tools. Each target protein was randomly shuffled to generate one entrapment protein[46]. Peptide C-termini were fixed during the shuffling. The code to generate the entrapment database and to calculate the false discovery proportion (FDP) can be found in "Code availability". The dataset published by Searle et al.[44] was used to search against the entrapment database. MaxQuant (version 2.4.13), MetaMorpheus (version 1.0.5),

MSFragger (version 4.1) in DDA mode, and MSFragger-DDA+ (version 4.1) were used. The parameters are the same as those used in the previous section. For MaxQuant, the "evidence.txt" and "proteinGroups.txt" files with decoys removed were used to count the peptide sequences and proteins, respectively. For MetaMorpheus, the "*_Peptides.psmtsv" and "*_ProteinGroups.tsv" files in the "Individual File Results" folder were used to count the peptide sequences and proteins, respectively. The decoys were removed, and peptides were filtered with "PEP_QValue < 0.01", and the proteins were filtered with "Protein_QValue < 0.01". For MSFragger and MSFragger-DDA+, the "peptide.tsv" and "combined_protein.tsv" files were used to count the peptide sequences and proteins, respectively. The "peptide.tsv" files contain the peptides filtered with 1% peptide- and protein-level FDR, and the "combined_protein.tsv" file contains the proteins filtered with 1% protein-level FDR. The raw parameter files, result files are available as described in "Data availability". The scripts to summarize the results and generate the figures are available as described in "Code availability".

### timsTOF DDA-PASEF data analysis

The dataset published by Wang et al.[47] was used to evaluate the performance of MSFragger-DDA+ when analyzing timsTOF DDA-PASEF data. There are two cell lines, A549 and K562. Each cell line has three biological replicates, and each biological replicate has four technical replicates. A Bruker timsTOF Pro mass spectrometer was used to generate the data. Details of the sample preparation and data acquisition can be found in the original publication. We used MSFragger (version 4.1) in DDA mode and MSFragger-DDA+ (version 4.1) to analyze the dataset. As in the previous section, FragPipe (version 22.0) was used to process the outputs of MSFragger and MSFragger-DDA+. In addition, IonQuant[35,48] was used to perform the label-free quantification with and without MBR, respectively. The "LFQ-MBR" workflow with MBR enabled and disabled was applied, respectively. The FASTA database is the Homo Sapiens proteome provided by the original publication[47] (20437 proteins including common contaminants from https://www.thegpm.org/crap/). Reversed decoy sequences were generated and appended to the target database. The maximum allowed missed cleavages were set to 2. The other parameters were the same as those described in the previous section. The "combined_protein.tsv" files were used to summarize the results and generate the figures. The "MaxLFQ Intensity" was used as the protein intensity. Detailed parameters, logs, and result files can be found in "Data availability".

### Wide-window acquisition data analysis

Datasets published by Truong et al.[9] and Matzinger et al.[10] were used to demonstrate the performance of MSFragger-DDA+ on WWA data. The first dataset contains 29 low-input samples with different isolation windows sizes (1.6, 2, 4, 8, 12, 18, 24, 48 Th), maximum injection time (54, 86, 118, and 246 ms), and MS/MS resolutions (30 K, 45 K, 60 K, and 120 K) from Truong et al[9]. Each sample has two technical replicates. There are also four high-input samples to be used as "library" during the MBR. The second dataset[9] contains four samples with different cell lines (K562 and HeLa) and gradient lengths (20 min and 40 min). Each sample contains eight technical replicates. Similar to the first dataset, there are also four high-input samples to be used as "library" runs. The third dataset[10] contains 43 samples from different isolation window sizes (1, 2, 3, 4, 5, 6, 7, 8, 12, 18, 24, 28, and 56 Th) and sample amounts (250 pg, 1 ng, 10 ng, 200 ng, and 400 ng). Each sample has three replicates. Thermo Orbitrap Exploris 480 mass spectrometers were used to generate these three datasets. Details of sample preparation and data acquisition can be found in the original publications. The data were analyzed using CHIMERYS[31] coupled with Proteome Discoverer 3.0, MetaMorpheus (version 1.0.5), and MSFragger-DDA+ (version 4.1) coupled with FragPipe (version 22.0). For MetaMorpheus and

MSFragger, the maximum allowed missed cleavages were set to 2. For MetaMorpheus, the precursor deconvolution was enabled to deconvolute co-fragmented peptides into separated spectra. The FASTA databases are from the original publications[9,10]. For FragPipe, the built-in "WWA" workflow was applied with the MBR settings adjusted accordingly. Detailed parameters, logs, and result files can be found in "Data availability".

### Phosphoproteome data analysis

To evaluate the performance of the phosphoproteome data, we used MSFragger (version 4.1) in DDA mode and MSFragger-DDA+ (version 4.1) to analyze the dataset published by Sacco et al.[49] There are six replicates of phosphorylation-enriched human samples. Details of the sample preparation and data acquisition can be found in the original publication. The FASTA database was downloaded from UniProt (20468 proteins including common contaminants). The reversed proteins were used as decoys. Similar to the previous experiments, FragPipe (version 22.0) integrated with MSBooster[37], Percolator[38], PTMProphet[40], ProteinProphet[41], Philosopher[42], and IonQuant[35,48] was used to perform the downstream analysis with the MSFragger and MSFragger-DDA+ results. FragPipe's "LFQ-phospho" workflow was used for the analysis. After filtering the results with global PSM- and protein-level 1% FDR, phosphosites with localization probability >0.75 were counted to evaluate the sensitivity. We also counted the number of phosphosequences after stripping the modifications from the peptide. Detailed parameters, logs, and result files can be found in "Data availability".

### Glioma data analysis

A Glioma study[50] dataset was used to demonstrate the performance of MSFragger-DDA+ on data from cancer proteomics experiments. There are 42 samples analyzed using the Thermo Q Exactive HF Orbitrap mass spectrometer. Details of the sample preparation and data acquisition can be found in the original publication. MSFragger (version 4.1) and MSFragger-DDA+ (version 4.1) were used to analyze the data. The outputs were processed using FragPipe (version 22.0). The FASTA database contains human reviewed proteins and common contaminants downloaded from UniProt (20468 proteins). The "LFQ-MBR" workflow was applied with "MBR top runs" set to 100. Details of the parameters and result files can be found as described in "Data availability". Section. FragPipe-Analyst[51] was used to perform downstream analysis using the output files generated by FragPipe.

### Runtime comparison

The dataset published by Searle et al.[44] was used to evaluate the speed of MSFragger-DDA+. MaxQuant (version 2.4.13), MetaMorpheus (version 1.0.5), and MSFragger (version 4.1) in DDA mode were also used. FragPipe (version 22.0) was used to process the outputs of MSFragger and MSFragger-DDA+. The mzML files converted[57] from the raw files were used in MetaMorpheus, MSFragger, and MSFragger-DDA+. MaxQuant was run using the raw files. All tools were run on a computer with Intel Xeon W-2235 CPU (3.80 GHz, 6 physical cores, 12 threads) and 128 GB RAM. The detailed log files can be found as described in "Data availability" section.

### Reporting summary

Further information on research design is available in the Nature Portfolio Reporting Summary linked to this article.

## Data availability

The raw MS/MS files used in this study can be found at the ProteomeXchange Consortium and PRIDE[58] partner repository or at the MassIVE[59] repository with the following accession codes: PXD027242, MSV000090552, PXD041421, PXD037527, PXD045500, PXD024427. The parameter, log, and result files generated in this study can be found at https://doi.org/10.5281/zenodo.13926175. Source data are provided with this paper.

## Code availability

The standalone version of MSFragger-DDA+ can be downloaded as part of MSFragger at https://msfragger.nesvilab.org/. FragPipe is available at https://github.com/Nesvilab/FragPipe. The program to generate the entrapment database in this study is publicly available and has been deposited in GitHub at https://github.com/Nesvilab/EntrapBench, under Apache 2.0 license. The code used to generate the figures in this study is publicly available and has been deposited in GitHub at https://github.com/Nesvilab/MSFragger-DDAPlus-Manuscript, under Apache 2.0 license.

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

## Acknowledgements

This work was supported in part by National Institutes of Health grants R01-GM-094231 (A.I.N.) and U24-CA271037 (A.I.N.). We thank Ryan Kelly and Thy Truong for the discussions regarding WWA data.

## Author contributions

F.Y. and A.I.N. developed the MSFragger-DDA+ algorithm. F.Y. implemented the algorithm in the software. F.Y., Y.D., and A.I.N. analyzed the results. F.Y. and A.I.N. wrote the manuscript with input from Y.D. A.I.N. and F.Y. conceived the study.

## Competing interests

A.I.N. and F.Y. receive royalties from the University of Michigan for the sale of MSFragger, IonQuant, and diaTracer software licenses to commercial entities. All license transactions are managed by the University of Michigan Innovation Partnerships office, and all proceeds are subject to university technology transfer policy. Other authors declare no competing interests.
