## [Peer Review file · Nature Communications]

MSFragger-DDA+ Enhances Peptide Identification Sensitivity with Full Isolation Window Search

Corresponding Author: Professor Alexey Nesvizhskii

Version 0:

Reviewer comments:

Reviewer #1

(Remarks to the Author)

The following paper introduces MSFragger-DDA+, a novel database search algorithm for liquid chromatography-mass spectrometry (LC-MS)-based proteomics. It addresses the limitations of traditional peptide identification tools by detecting co-fragmented peptides and improving the sensitivity, accuracy, and comprehensiveness of proteomics data analysis.

Key features of this paper are:

1. Improved Peptide Identification Sensitivity: Detects co-fragmented peptides with high sensitivity and speed.
2. The stringent False Discovery Rate (FDR) control ensures accurate peptide identification through robust feature detection and filtering.
3. Wide-Window Acquisition (WWA) Support: This feature is suitable for WWA data, enabling the analysis of complex proteomics samples.
4. The system utilizes MSFragger's fragment ion indexing algorithm for efficient and accurate peptide identification.

Comments

Introduction

1. Consider adding a brief statement on the significance of proteomics research and its applications.
2. Provide a clearer transition between the introduction of MSFragger-DDA+ and the description of its features.
3. Consider adding a sentence or two to outline the structure of the paper.
4. While you mention the advantages of WWA DDA, it might be helpful to briefly explain what "wide-window acquisition" entails for readers unfamiliar with the term.

Results

Suggestions for Improvement:

1. Provide more details on the greedy algorithm used for shared fragment removal.
2. Clarify the advantages of the targeted XIC extraction approach compared to untargeted methods.

Major Comments:

1. Consider providing more detailed information on the statistical significance of the performance differences between MSFragger-DDA+ and other tools.
2. Please explain how you applied the 1% FDR filtering and whether you used the same filtering criteria for all tools.
3. It would be helpful to include additional performance metrics, such as precision, recall, and area under the receiver operating characteristic curve (AUC-ROC).
4. Please clarify how the peptide-level FDP estimates (Supplementary Figure 1) relate to the protein-level FDP estimates (Figure 2d).
5. Consider discussing the implications of MSFragger-DDA+'s performance on WWA data for single-cell proteomics applications.
6. Consider discussing the clinical implications of the identified differentially expressed proteins and pathways.
7. Discuss potential limitations and challenges for MSFragger-DDA+.
8. Please clarify how MSFragger-DDA+ handles cases with high levels of noise or contamination.
9. It would be helpful to provide more details on the computational resources required for MSFragger-DDA+ in the paper.

Minor Comments:

1. Consider rephrasing "hyperscore" to provide clearer context for readers unfamiliar with this term.
2. Typo: "FragPipe (version 22.0)" (line 150) should include a citation or reference.
3. I would suggest using more common terminology, such as "high-throughput," in lines 38–39 of the high-throughput method.

4. Typo: "entrapment sequence" (line 172) could be rephrased for clarity.

Questions:

1. How does MSFragger-DDA+ handle cases where multiple peptides have identical or very similar fragmentation patterns?
2. How does the choice of fragmentation method (CID vs. HCD) impact MSFragger-DDA+ performance?

(Remarks on code availability)

<https://github.com/Nesvilab/MSFragger-DDAPlus-Manuscript>

Here, the author could have a better readme file for readers with the title and the author's list of the paper.

Every script lacks a description and a purpose statement. Additionally, the data files for the figure generation script are absent.

The link contains a number of scripts without any data files, and it is unclear what it is. I believe every script also needs proper documentation.

<https://github.com/Nesvilab/EntrapBench>, <https://github.com/Nesvilab/FragPipe>

These two links are functional, have a clear explanation of every aspect, and were able to install and run the basic utility of the package.

Reviewer #2

(Remarks to the Author)

The manuscript from Yu et al presents the authors' development of a database search algorithm, MSFragger-DDA+, that was designed from the ground-up to identify multiple peptides per fragmentation spectrum. The occurrence of so-called chimeric spectra is well-known and if not considered, can be a major source of reduced peptide identification rates. As the authors note, some search tools do not have the ability to identify > 1 peptide per spectrum, while a few tools have made progress on this issue. The authors concisely describe in the Introduction the strategies that previous tools used to identify multiple peptides per fragmentation event. The authors draw on several publicly available datasets to demonstrate the performance characteristics of their new DDA+ algorithm. Overall, their computational analyses consistently show a huge peptide identification benefit of up to 50% using DDA+ versus the original MSFragger (DDA) algorithm. This translates into more proteins quantified and increased consistency among biological replicates, which enables the identification of more differential targets compared to the original study's computational pipeline. Critically, the additional sensitivity did not sacrifice false positive identifications as the estimated FDRs were maintained to acceptable levels.

Overall, the manuscript is well organized, clearly written, and contains no critical experimental flaws. I only want to raise one primary and one minor issue with the submitted manuscript.

1) The identification of posttranslational modifications is also provided as a core feature of database search algorithms. Often, they are identified by first performing an enrichment step to reduce sample complexity. It would be important for the authors to show in this type of sample, whether identification of chimeric spectra improves identification rates of modified peptides and modified sites with high confidence. The authors platform, FragPipe, contains PTM site localization tools and therefore it may be relatively straightforward to include this analysis, as well as comparison at least one other tool.

2) The authors state in the introduction on lines 74-78 that most strategies rely on the MS1 peaks and it may not be possible for all co-fragmented peptides to have detectable (MS1) precursor peaks. This section implies that FragPipe can identify additional peptides without requiring the precursor to be detected, however, looking at Figure 1, it shows all fragment ions being matched to precursor peaks, and the associated text states (line 129) that PSM with lower-quality XICs are discarded. Please clarify.

(Remarks on code availability)

I have successfully run the FragPipe software including the DDA+ algorithm. I found similar peptide and protein identification performance increases with my own test dataset.

Version 1:

Reviewer comments:

Reviewer #1

(Remarks to the Author)

I am pleased to report that the authors have thoroughly addressed all the concerns I mentioned in my previous review. They have provided clear and convincing responses to my comments, and the revised manuscript demonstrates significant improvements.

After carefully reviewing the revised manuscript, I am satisfied with the current state of the paper. The authors have done an excellent job in revising and strengthening their work. The manuscript is now well-written, clearly organized, and effectively communicates its key findings.

(Remarks on code availability)

The authors created a well-organized and useful codebase that supports the research in the manuscript well. The code is

well documented.

Reviewer #2

(Remarks to the Author)

The authors have addressed all my comments.

(Remarks on code availability)

Reviewer #1 (Remarks to the Author):

The following paper introduces MSFragger-DDA+, a novel database search algorithm for liquid chromatography-mass spectrometry (LC-MS)-based proteomics. It addresses the limitations of traditional peptide identification tools by detecting co-fragmented peptides and improving the sensitivity, accuracy, and comprehensiveness of proteomics data analysis.

Key features of this paper are:

1. Improved Peptide Identification Sensitivity: Detects co-fragmented peptides with high sensitivity and speed.
2. The stringent False Discovery Rate (FDR) control ensures accurate peptide identification through robust feature detection and filtering.
3. Wide-Window Acquisition (WWA) Support: This feature is suitable for WWA data, enabling the analysis of complex proteomics samples.
4. The system utilizes MSFragger's fragment ion indexing algorithm for efficient and accurate peptide identification.

Response: We thank the reviewer for the summary and positive feedback. We have addressed all the comments. Please find our point-by-point responses below. Our responses are presented in bold font, and quoted sentences are formatted in italic bold font.

Comments

Introduction

1. Consider adding a brief statement on the significance of proteomics research and its applications.

Response: We thank the reviewer for the suggestion. We have added the following sentence in the Introduction section.

It has been used to study post-translational modifications¹ (PTMs), cancer diseases^{2, 3}, SARS-Cov-2⁴, clinical plasma samples⁵, protein-protein interactions⁶, etc.

2. Provide a clearer transition between the introduction of MSFragger-DDA+ and the description of its features.

Response: We have revised the following sentences in the Introduction section.

Here, we further extend this strategy and present MSFragger-DDA+, a new search mode of MSFragger. It utilizes high-resolution DDA MS2 to detect co-fragmented

peptides with high sensitivity and accuracy. Unlike other DDA tools, this method does not perform MS1 detection or spectral deconvolution before the database search.

3. Consider adding a sentence or two to outline the structure of the paper.

Response: We have added the following sentences in the Introduction section.

In the following sections, we first provide an overview of the MSFragger-DDA+ algorithm and its integration into the FragPipe computational platform. Next, we evaluate its performance across various datasets, comparing its sensitivity, speed, and accuracy with existing methods. We conclude by discussing the implications of the MSFragger-DDA+ approach and related topics. Finally, the MSFragger-DDA+ algorithm and the experiments' parameters are detailed in the Methods section.

4. While you mention the advantages of WWA DDA, it might be helpful to briefly explain what “wide-window acquisition” entails for readers unfamiliar with the term.

Response: We have revised the following sentences in the Introduction section.

The WWA method generates MS2 spectra similar to those in DDA but employs wider isolation windows, allowing for reduced duty cycles. This approach allocates more time for ion accumulation, enabling the production of high-quality tandem mass spectra, particularly beneficial for low-input samples.

Results

Suggestions for Improvement:

1. Provide more details on the greedy algorithm used for shared fragment removal.

Response: We thank the reviewer for the suggestion. We have added more details about the greedy algorithm.

Given a list of PSMs derived from the same tandem mass spectrum, the greedy algorithm operates iteratively. First, it removes the fragment peaks matched to the top-scoring PSM. Next, it eliminates the top-scoring PSM from the list, re-calculates the hyperscores for the remaining PSMs, and re-ranks them accordingly. These steps are repeated until there are insufficient fragment peaks to match any PSM.

2. Clarify the advantages of the targeted XIC extraction approach compared to untargeted methods.

Response: We have revised the following sentences in the Results section to provide more details.

In contrast to the “untargeted” MS feature detection approach^{22, 28, 36} that may struggle to accurately detect and extract low-abundance features and their peak curves, the targeted approach reliably extracts the XICs given the theoretical m/z values. This extraction is restricted to the retention time range associated with the corresponding MS2 scan, ensuring high specificity and accuracy. Moreover, the targeted approach does not suffer from the challenges of untargeted deconvolution of overlapped isotopic peak clusters which might result in incorrect charge and m/z determination. Thus, the targeted XIC detection is more sensitive, even when precursor XICs are of low quality. It maximizes the potential to identify co-fragmented peptides with low-abundance precursor signals.

Major Comments:

1. Consider providing more detailed information on the statistical significance of the performance differences between MSFragger-DDA+ and other tools.

Response: The differences between MSFragger-DDA+ and other tools are obvious in Figure 2 and 4. We looked for an example of other manuscripts that try to apply a statistical test for the numbers of identifications and runtime, but did not find a good example applied when the improvements were as obvious as in our case.

2. Please explain how you applied the 1% FDR filtering and whether you used the same filtering criteria for all tools.

Response: The FDR filtering was conducted by the tools themselves. To ensure fair comparisons, we made every effort to align the FDR filtering parameters across the tools as closely as possible. The following descriptions can be found in the Methods section. Detailed parameters files and log files can be found in the Data availability section.

The decoys were removed, and the peptides were filtered with “PEP_QValue < 0.01”. We used the “peptide.tsv” files, which were filtered at 1% peptide- and protein-level FDR, to count the peptide sequences.

The decoys were removed, and peptides were filtered with “PEP_QValue < 0.01”, and the proteins were filtered with “Protein QValue < 0.01”. For MSFragger and MSFragger-DDA+, the “peptide.tsv” and “combined_protein.tsv” files were used to count the peptide sequences and proteins, respectively. The “peptide.tsv” files contain the peptides filtered with 1% peptide- and protein-level FDR, and the “combined_protein.tsv” file contains the proteins filtered with 1% protein-level FDR.

3. It would be helpful to include additional performance metrics, such as precision, recall, and area under the receiver operating characteristic curve (AUC-ROC).

Response: Calculating these metrics requires knowing the ground truth to count the true positives, false positives, true negatives, and false negatives, most of which are not feasible in this manuscript.

4. Please clarify how the peptide-level FDP estimates (Supplementary Figure 1) relate to the protein-level FDP estimates (Figure 2d).

Response: They are two different levels of FDP. That was why we calculated the FDP for both. The following explanation can be found from the Methods section.

5. Consider discussing the implications of MSFragger-DDA+'s performance on WWA data for single-cell proteomics applications.

Response: We thank the reviewer for the suggestion. We have added the following in the discussion section regarding the implications of MSFragger-DDA+'s performance on WWA single-cell data in the Discussion section.

The application of MSFragger-DDA+ to WWA data has significant implications in single-cell proteomics. Single-cell proteomic studies often require the analysis of extremely low-input samples, whereas traditional DDA approaches may struggle with peptide identification because of limited ion signal strength and suboptimal fragmentation quality. By utilizing an enlarged isolation window size, the duty cycle is reduced, allowing more time for ion accumulation. This ensures sufficient ions to generate abundant fragmented peaks, resulting in high-quality yet chimeric tandem mass spectra. MSFragger-DDA+'s capability to sensitively identify co-fragmented peptides from these chimeric spectra significantly expands the depth of proteome coverage achievable in single-cell experiments. Moreover, MSFragger-DDA+ is substantially faster than the existing tools that support WWA data, an advantage that is particularly crucial for single-cell studies involving large datasets with extensive analysis requirements. This speed enhancement addresses a critical bottleneck in data processing, thereby enabling more practical and high-throughput single-cell proteomics workflows that use WWA method.

6. Consider discussing the clinical implications of the identified differentially expressed proteins and pathways.

Response: In the manuscript, we use MSFragger-DDA+ and traditional MSFragger to analyze the clinical glioma dataset. We demonstrated the advancement of MSFragger-DDA+ over the traditional MSFragger. We also showed that the results of MSFragger-DDA+ and the traditional MSFragger have large overlaps, which indicates the correctness and robustness of MSFragger-DDA+. The discussion about the clinical implications is beyond the scope of this manuscript that focuses on the computational method development.

7. Discuss potential limitations and challenges for MSFragger-DDA+.

Response: We thank the reviewer for the suggestion. We have added discussion about the limitations and challenges.

In scenarios involving large search spaces, such as non-specific searches for human leukocyte antigen peptides, the full isolation window approach may present computational challenges and result in reduced sensitivity owing to the extensive number of candidate peptides. Furthermore, the current version of MSFragger-DDA+ does not support open or mass-offset searches, thereby limiting its applicability in the identification and discovery of unknown PTMs.

8. Please clarify how MSFragger-DDA+ handles cases with high levels of noise or contamination.

Response: It performs the XIC detection and filtering after the database search to remove noisy peaks. To account for the contaminants, we always add common contaminants to the protein database during the search. The details can be found in the Results and Methods sections.

9. It would be helpful to provide more details on the computational resources required for MSFragger-DDA+ in the paper.

Response: We thank the reviewer for the suggestion. We have added the information to the Results section.

MSFragger-DDA+ has been highly optimized for fast processing speeds and efficient memory usage. For most datasets, it can complete a tryptic database search on a standard desktop computer equipped with 32 GB of memory and a mainstream CPU. For tasks such as phosphopeptide or non-specific digested peptide identification, a larger memory size, such as 64-128 GB, is recommended.

Minor Comments:

1. Consider rephrasing "hyperscore" to provide clearer context for readers unfamiliar with this term.

Response: This terminology has been widely used in the proteomics community for a long time and has been described in numerous publications. We have also cited two references regarding the hyperscore. Therefore, we do not believe it is necessary to rephrase it.

Hyperscore^{11, 33} is used to measure the peptide-spectrum similarity during this process.

2. Typo: "FragPipe (version 22.0)" (line 150) should include a citation or reference.

Response: We have cited the individual tools in FragPipe.

To make MSFragger-DDA+ easy to access and user-friendly, we have integrated it into FragPipe computational suite. The output can be seamlessly processed by FragPipe modules, including MSBooster³⁷ deep-learning-based rescoring, Percolator³⁸ PSM re-ranking, PeptideProphet³⁹ PSM rescoring, PTMProphet⁴⁰ modification localization, ProteinProphet⁴¹ protein grouping, Philosopher⁴² false discovery rate (FDR) filtering, IonQuant³⁵ quantification, EasyPQP spectral library generation, PDV⁴³ and Skyline²⁵ visualization.

3. I would suggest using more common terminology, such as "high-throughput," in lines 38–39 of the high-throughput method.

Response: We apologize for the oversight. We have corrected it.

4. Typo: "entrapment sequence" (line 172) could be rephrased for clarity.

Response: We have added more explanations.

The entrapment database contains the entrapment sequences that do not exist in the original database. Given the original target database, we generated an entrapment sequence for each target protein sequence by random shuffling. During the shuffling, the peptide C-termini were fixed.

Questions:

1. How does MSFragger-DDA+ handle cases where multiple peptides have identical or very similar fragmentation patterns?

Response: MSFragger-DDA+ uses the greedy algorithm to remove the fragments shared by multiple peptides to avoid "double counting". Details can be found in the Results and Methods sections.

Then, MSFragger-DDA+ rescues the PSMs by removing the fragments shared by multiple peptides using a greedy algorithm²⁹. Given a list of PSMs derived from the same tandem mass spectrum, the greedy algorithm operates iteratively. First, it removes the fragment peaks matched to the top-scoring PSM. Next, it eliminates the top-scoring PSM from the list, re-calculates the hyperscores for the remaining PSMs, and re-ranks them accordingly. These steps are repeated until there are insufficient fragment peaks to match any PSM.

2. How does the choice of fragmentation method (CID vs. HCD) impact MSFragger-DDA+ performance?

Response: As far as we know, there is no significant differences in the performance.

Reviewer #1 (Remarks on code availability):

<https://github.com/Nesvilab/MSFragger-DDAPlus-Manuscript>

Here, the author could have a better readme file for readers with the title and the author's list of the paper.

Every script lacks a description and a purpose statement. Additionally, the data files for the figure generation script are absent.

The link contains a number of scripts without any data files, and it is unclear what it is. I believe every script also needs proper documentation.

Response: We have added more information to the README of the GitHub repository. The scripts also have concise comments to describe the purpose of the code.

The data files have already been uploaded to Zenodo as described in the Data availability section.

<https://github.com/Nesvilab/EntrapBench>, <https://github.com/Nesvilab/FragPipe>

These two links are functional, have a clear explanation of every aspect, and were able to install and run the basic utility of the package.

Response: We thank the reviewer for the positive feedback.

Reviewer #2 (Remarks to the Author):

The manuscript from Yu et al presents the authors' development of a database search algorithm, MSFragger-DDA+, that was designed from the ground-up to identify multiple peptides per fragmentation spectrum. The occurrence of so-called chimeric spectra is well-known and if not considered, can be a major source of reduced peptide identification rates. As the authors note, some search tools do not have the ability to identify > 1 peptide per spectrum, while a few tools have made progress on this issue. The authors concisely describe in the Introduction the strategies that previous tools used to identify multiple peptides per fragmentation event. The authors draw on several publicly available datasets to demonstrate the performance characteristics of their new DDA+ algorithm. Overall, their computational analyses consistently show a huge peptide identification benefit of up to 50% using DDA+ versus the original MSFragger (DDA) algorithm. This translates into

more proteins quantified and increased consistency among biological replicates, which enables the identification of more differential targets compared to the original study's computational pipeline. Critically, the additional sensitivity did not sacrifice false positive identifications as the estimated FDRs were maintained to acceptable levels.

Overall, the manuscript is well organized, clearly written, and contains no critical experimental flaws. I only want to raise one primary and one minor issue with the submitted manuscript.

Response: We thank the reviewer for the compressive summary and positive feedback. Please find our point-to-point response to the comments in the following. Our responses are presented in bold font, and quoted sentences are formatted in italic bold font.

1) The identification of posttranslational modifications is also provided as a core feature of database search algorithms. Often, they are identified by first performing an enrichment step to reduce sample complexity. It would be important for the authors to show in this type of sample, whether identification of chimeric spectra improves identification rates of modified peptides and modified sites with high confidence. The authors platform, FragPipe, contains PTM site localization tools and therefore it may be relatively straightforward to include this analysis, as well as comparison at least one other tool.

Response: We thank the reviewer for the suggestion. We have added one experiment with phosphoproteome data in the Results and Methods sections.

MSFragger-DDA+ has higher sensitivity in analyzing phosphoproteome data

To demonstrate the enhanced capability of MSFragger-DDA+ in identifying peptides from phosphorylation-enriched data, we benchmarked MSFragger-DDA+ against MSFragger in DDA mode using the dataset published by Sacco et al⁴⁹ (Methods). The dataset includes six replicates of a human sample. MSFragger-DDA+ identified 22378 phosphosites, compared to 21249 identified by MSFragger in DDA mode, with an overlap of 20018 phosphosites between the two modes. After stripping modifications, MSFragger-DDA+ identified 23948 phosphosequences, whereas MSFragger in DDA mode identified 21070, with 19907 overlapping phosphosequences. These results demonstrate that MSFragger-DDA+ identified ~5% more phosphosites and ~14% more phosphosequences than the traditional DDA mode. The improvement is less pronounced compared to the results from proteome data. This difference is likely due to the larger search space associated with phosphoproteome peptide identification, which reduces the sensitivity under the same FDR control. Additionally, we applied a 0.75 localization probability cut-off for phosphosites. Since co-fragmented peptides normally have lower fragmentation quality and localization confidence, this cut-off further narrowed the difference between the DDA+ and DDA modes.

Phosphoproteome data analysis

To evaluate the performance of the phosphoproteome data, we used MSFragger (version 4.1) in DDA mode and MSFragger-DDA+ (version 4.1) to analyze the dataset published by Sacco et al⁴⁹. There are six replicates of phosphorylation-enriched human samples. Details of the sample preparation and data acquisition can be found in the original publication. The FASTA database was downloaded from UniProt (20468 proteins including common contaminants). The reversed proteins were used as decoys. Similar to the previous experiments, FragPipe (version 22.0) integrated with MSBooster³⁷, Percolator³⁸, PTMProphet⁴⁰, ProteinProphet⁴¹, Philosopher⁵⁷, and IonQuant³⁵, 48 was used to perform the downstream analysis with the MSFragger and MSFragger-DDA+ results. FragPipe's "LFQ-phospho" workflow was used for the analysis. After filtering the results with global PSM- and protein-level 1% FDR, phosphosites with localization probability >0.75 were counted to evaluate the sensitivity. We also counted the number of phosphosequences after stripping the modifications from the peptide. Detailed parameters, logs, and result files can be found in Data availability.

2) The authors state in the introduction on lines 74-78 that most strategies rely on the MS1 peaks and it may not be possible for all co-fragmented peptides to have detectable (MS1) precursor peaks. This section implies that FragPipe can identify additional peptides without requiring the precursor to be detected, however, looking at Figure 1, it shows all fragment ions being matched to precursor peaks, and the associated text states (line 129) that PSM with lower-quality XICs are discarded. Please clarify.

Response: We apologize for the confusion. Our method has a lower requirement than the other methods. Both high- and low-quality precursor XICs can be detected by our method. We have added the following clarification to the Introduction section.

MSFragger-DDA+ has a lower requirement for the quality of precursor XICs, enabling both high- and low-quality XICs to be detected and utilized for refining PSM scores.

Reviewer #2 (Remarks on code availability):

I have successfully run the FragPipe software including the DDA+ algorithm. I found similar peptide and protein identification performance increases with my own test dataset.

Response: We thank the reviewer for the testing and positive feedback.